# *Moringa oleifera* Oil Nutritional and Safety Impact on Deep-Fried Potatoes

**DOI:** 10.3390/foods12244416

**Published:** 2023-12-08

**Authors:** Silia Boukandoul, Farid Zaidi, Carla S. P. Santos, Susana Casal

**Affiliations:** 1Département des Sciences Alimentaires, Faculté des Sciences de la Natureet de la Vie, Université de Bejaia, Route Targa Ouzemour, Bejaia 06000, Algeria; 2REQUIMTE/LAQV, Laboratory of Bromatology and Hydrology, Faculty of Pharmacy, Porto University, Rua de Jorge Viterbo Ferreira, 228, 4050-313 Porto, Portugalsucasal@ff.up.pt (S.C.); 3Département de Biochimie-Microbiologie, Faculté des Sciences Biologiques et des Sciences Agronomiques, Université Mouloud Mammeri de Tizi-Ouzou, Tizi-Ouzou 15000, Algeria

**Keywords:** deep-fried potatoes, chemical composition, antioxidant activity, acrylamide, volatiles

## Abstract

Aiming to understand the nutritional impact of *Moringa oleifera* oil (MOO) on the quality of fried potatoes as consumed, a frying study using intermittent frying at 180 °C was conducted over 5 days, with a total heating time of 15 h, against olive (OO) and sunflower (SFO) oils. Additionally, due to MOO’s higher costs, a SFO/MOO blend (80/20 *w*/*w*) was tested. With similar fat incorporation and moisture contents, potato lipid composition revealed the impact of oil oxidation over the frying time, gradually decreasing the content of unsaturated fatty acids and antioxidants, including vitamin E, carotenoids and ascorbic acid, and increasing the incorporation of *trans* fatty acids (TFAs) and volatile aldehydes. When the potatoes fried at the ninth hour of heating are compared, MOO and OO were still able to protect potato ascorbic acid better than SFO, due to the low oxidative stress imposed by their fatty-acid composition. SFO, on the contrary, with linoleic acid as the main fatty acid, and despite its higher content of vitamin E, demonstrated higher oxidative stress and increased incorporation of alkenals and alkadienals. Acrylamide content was generally low, as were the *trans* fatty acids formed and incorporated with frying time, with MOO fried potatoes having lower amounts of all these process contaminants. Interestingly, the blend SFO/MOO (80/20 *w*/*w*) doubled the amount of vitamin E in fried potatoes when compared with SFO alone, increased the ascorbic acid protection and reduced by half the amounts of volatile aldehydes, indicative of an efficient reduction of the oxidative status of the SFO-fried potatoes, with benefits to the consumer from a health point of view.

## 1. Introduction

Fried potatoes are eaten all around the world. The potato’s nutritional relevance is supported by its richness in carbohydrates, minerals, proteins, dietary fiber, vitamins (such as vitamin C and pro-vitamin A) and phytochemicals, particularly phenolic compounds [1]. However, when fried, the oil from the frying medium becomes part of the fried food. Therefore, the nutritional quality of fried potatoes is the combined result of the initial potato constituents, the incorporated frying oil and the products of their physical and chemical interactions [2]. With the heated frying oil becoming part of the food, its characteristics are of utmost importance to the fried food’s quality and safety. Frying oils can supply additional nutritional value to fried foods, as a result of the incorporation of important lipid components, such as essential fatty acids and vitamin E, increasing also their energetic density [2]. Therefore, it is of crucial importance to choose vegetable oils with adequate thermal resistance for deep-frying, reducing the probability of producing oxidized lipids and several other degradation compounds that can compromise the quality of the potatoes and ultimately human health.

Different types of oils and fats are available worldwide and widely accepted for deep-frying purposes, including soybean, palm, groundnut, canola, sunflower and olive oils. During the frying process, diverse chemical reactions occur, such as oxidation, hydrolysis and polymerization [3,4], that affect the performance and quality of the oil. However, under the high temperatures of the frying process, some compounds are degraded, and undesirable toxic molecules may be generated, either in the frying oil, as *trans* fatty acids or volatile aldehydes, etc., or in the fried food, namely, Maillard reaction products, acrylamide, etc., whose intake should be avoided or at least limited [2,5].

The most common vegetable oils worldwide have a polyunsaturated nature, such as soybean or sunflower oil, with an inherent increased susceptibility to oxidation under thermal stress [6]. High-oleic vegetable oils, on the other hand, are increasingly being recognized as appropriate for frying due to their fatty-acid profile and oxidative stability [6,7]. *Moringa oleifera* oil (MOO) is a high-oleic vegetable oil typical of the Mediterranean region, recognized for its nutritional, technological and sensorial properties. Its use for frying, alone or in blends, is reported in the literature, including the physical and chemical alterations of the oil bath under deep-frying conditions, as well as the sensory characteristics of fried potatoes [6,8,9]. However, the compositional and nutritional quality of potatoes fried in MOO, alone or blended with other common vegetable oils through frying time, has been given limited attention. 

In this context, this study was designed to simulate domestic deep-fat frying of potatoes using MOO. Aware of its high cost, and following previous positive achievements in delaying sunflower oil oxidation when blended with SFO [9], the study included MOO and SFO alone, and a blend with 80% sunflower oil (SFO/MOO, 80/20, *v*/*v*). The study simulates the thermal stress induced by the preparation of two daily meals over five days, without repositioning of the oil, corresponding to a daily accumulation of 3 h of heating, in a total of 15 h. Plain OO, a high-oleic oil also from the Mediterranean region, and sunflower oil (SFO), one of the commonest polyunsaturated oils in the same region, were used as references. The study focused on the physicochemical and nutritional changes of the fried potatoes, including an evaluation of the potato’s antioxidant activity, acrylamide and volatile compounds.

## 2. Materials and Methods

### 2.1. Reagents and Standards

All reagents were of analytical or chromatographic grade and were supplied from Merck (Darmstadt, Germany) or Sigma–Aldrich (St. Louis, MO, USA). A certified fatty acid methyl esters standard mixture (CRM47885) from Supelco (Bellefonte, PA, USA) was used. Gallic acid, monostearin and 1,2,3-trichloropropane were from Sigma-Aldrich. The internal standard for vitamin E analysis (tocol) was supplied by Matreya, Inc. (Bellefonte, PA, USA). All experiments and analytical determinations were performed in triplicate.

### 2.2. Samples Preparation 

Red potatoes (*Solanum tuberosum* L., Mozart variety) were chosen for the assay due to their frying aptitude and availability in local supermarkets (Porto, Portugal). Every frying day, potatoes were freshly washed, hand-peeled and cut into cubes of 1 cm, then washed with tap water and drained to reduce moisture before being used for frying tests. Four different oils were used in this study; *Moringa oleifera* oil (MOO) was obtained from Moringa seeds cultivated in Tamanrasset, Algeria, after oil extraction and the degumming process as detailed in Boukandoul et al. [9]. Sunflower oil and olive oil (a blend of refined olive oil and virgin olive oil) were acquired in local supermarkets. The blend of sunflower oil with *Moringa oleifera* oil was prepared at 80:20 (*w*/*w*) proportions.

### 2.3. Frying Process

Fresh potato cubes were intermittently deep-fried at 180 ± 5 °C at a potato/oil ratio of 1:10, using domestic electric deep-fryers (600 W, 0.5 L; Mexxtronics, Frankfurt, Germany). A digital thermometer was used to control the temperature during the frying process. During five consecutive days, each oil was heated twice a day for 90 min, separated by 3 h of cooling, followed by overnight cooling (18 h). In each heating session of 90 min, two batches of potato cubes were fried at minute 30 and minute 60, for 3 min each. This frying protocol was designed to simulate domestic deep-fatfrying of two daily meals, without repositioning of the oil over five days, corresponding to a cumulative heating time of 3, 6, 9, 12 and 15 h. The fried potatoes were immediately analyzed for color and then mixed and homogenized in a food blender. A portion was used for immediate moisture and ascorbic acid determinations, and the remainder was stored at −20 °C until used for other analysis. Raw potatoes were also taken for analysis. The oils were preserved for total polar compound (TPC) analysis.

### 2.4. Physicochemical Analysis

#### 2.4.1. Total Polar Compounds 

The TPCs of the vegetable oils were analyzed by high-performance size-exclusion chromatography (HPLC) as detailed by Marquez-Ruiz et al. [10]. Polar compounds were submitted to a separation from triglycerides on a silica solid-phase column (1 g, Tecnocroma, Barcelona, Spain), using monostearin (1-stearoyl-rac-glycerol, Sigma-Aldrich, St. Louis, MI, USA) as an internal standard. Jasco HPLC equipment (Jasco, Tokyo, Japan) with refractive index detection (Gilson, 132 model, Villiers-le-Bel, France) was used, equipped with a Phenomenex column (Phenogel, 100 Å, 600 × 7.8 mm ID, 5 μm film thickness Alcobendas, Spain), and with Tetrahydrofuran (THF, Merck, Darmstadt, Germany) as the mobile phase at a flow rate of 1 mL min^−1^. Quantification of polar compounds (PC) was based on the internal standard amounts, with a quantification limit of 2 µg per injection. 

#### 2.4.2. Color Coordinates

The color coordinates of fried potatoes were measured directly after sampling in six different locations. A Konica Minolta colorimeter (Chroma Meter CR-400, Osaka, Japan) with illuminant D65 was used. The color space system used was CIE-L*a*b* to measure color coordinate values; the L* value represents the lightness–darkness dimension (0 to 100), the a* value represents the red–green dimension (−120 to 120) and the b* value represents the yellow–blue dimension (−120 to −120). Total color differences between raw and fried potatoes, labeled ΔE, were calculated according to the equation ΔE = $\sqrt{{\Delta L*}^{2}+{\Delta a*}^{2}+{\;\Delta b*}^{2}}$ [11].

#### 2.4.3. Moisture

Moisture was determined by drying at 105 °C until reaching a constant weight and expressed in g 100 g^−1^ of raw or fried potato sample.

#### 2.4.4. Ascorbic Acid

Total ascorbic acid was analyzed by HPLC following the method described by Santos et al. [12], after extraction with tris-(2-carboxy-ethyl)-phosphine-hydrochloride (2.5 mM), 3% of metaphosphoric acid and 8% of acetic acid. Chromatographic separation was achieved using a YMC-Trial Diol Hilic column (150 × 3.0 mm, 3 µm, Ireland) with a gradient of acetate buffer (pH = 5) and acetonitrile (10:90, *v*/*v*) at a flow rate of 0.8 mL min^−1^, and detection at 266 nm. Quantification was based on external L-ascorbic acid calibration curves, with standard solutions subjected to the same extraction procedure as samples, and the results were expressed in mg L-ascorbic acid 100 g^−1^ of fresh or fried potatoes.

#### 2.4.5. Lipid Content

The lipid content of deep-fried potatoes was determined by Soxhlet extraction using petroleum ether (40–60 °C) and expressed in g 100 g^−1^ of fried potatoes.

#### 2.4.6. Fatty Acids and Vitamin E

To avoid the degradation of potato lipids, particularly liposoluble vitamins, cold-fat extraction was adopted. Briefly, 10 g of homogenized fried potatoes were extracted overnight under refrigeration with adequate amounts of 2-propanol and cyclohexane, in the presence of butylated hydroxytoluene (BHT) and ascorbic acid. After the separation of the cyclohexane upper phase with NaCl (1%) solution, the lipids were recovered after evaporation of solvent remains under a nitrogen stream and stored at 4 °C until being used for fatty acids and vitamin E analysis.

For fatty-acid composition, the glycerides were converted to their methyl esters (FAME) by alkaline transesterification [13] and determined using an Agilent 7890A gas chromatograph with a flame ionization detector (FID) (Palo Alto, Santa Clara, CA, USA, equipped with a Select FAME column (50 m × 0.25 mm i.d.; Agilent J&W), with helium as a carrier gas at a flow rate of 1 mL min^−1^. Injection (1 µL) was made in split mode (1:50) at 260 °C, with a total run time of 45 min. Each FAME was calculated on a mass basis using the internal standard amount (triundecanoin), after proper calibration on the FID response for each fatty acid with a certified standard mixture, and results were expressed in g 100 g^−1^ of fresh or fried potatoes. 

Tocopherols and tocotrienols of potato lipid were analyzed by HPLC [14] with fluorescence detection (Jasco, Japan), after a direct dilution of lipid extracts in n-hexane in the presence of tocol (Matreya, Oklahoma City, OK, USA) as an internal standard. The separation was performed on a normal-phase silica column (SupelcosilTM LC-SI; 7.5 cm × 3 mm; 3 mm) (Supelco, Bellefonte, PA, USA), conditioned at 25 °C (ECOM, ECO 2000, Praha, Czech Republic) and eluted with a mobile phase of 1,4-dioxane in n-hexane (2.5%, *v*/*v*), at a flow rate of 0.75 mL min^−1^. Vitamin E compounds were identified and quantified using individual calibration curves of tocopherols and tocotrienols standards, and concentrations were expressed in µg 100 g^−1^ of fresh or fried potatoes.

#### 2.4.7. Total Carotenoids 

Total carotenoids of fresh and fried potatoes were determined following the method described by Nagata and Yamashita [15], after extraction with acetone–hexane mixture (4:6, *v*/*v*) and based on the readings at 663, 645, 505 and 453 nm using a SPECTROstar Nano spectrophotometer (BMG LABTECH GmbH, Offenburg, Germany). Total carotenoids were estimated following the equation given by the authors and expressed in µg of β-carotene equivalents 100 g^−1^ of samples.

#### 2.4.8. Phenolic Compounds and Antioxidant Activity

Phenolic compounds of fresh and fried potatoes were extracted twice with methanol/water mixtures (50:50 followed by 70:30 (*v*/*v*); pH = 2). Both supernatants were recovered after centrifugation, combined and used to determine total phenolic compounds and antioxidant capacity [16]. 

For total phenolic compounds, a volume of the extract was mixed with Folin–Ciocalteu reagent (10%, H_2_O) and sodium carbonate solution 7.5% (*w*/*v*), then left to react in the dark for 30 min. UV readings were performed at 765 nm, and the results were expressed in mg of gallic acid equivalents 100 g^−1^ of fresh or fried potatoes.

The determination of the total antioxidant activity was carried out by measuring spectrophotometrically at 515 nm the disappearance of 2,2-diphenyl-1-picrylhydrasylfree radicals (DPPH°) after reaction with the extracts according to Espin et al. [17]. The results were recorded in mg of gallic acid equivalents 100 g^−1^ of fresh or fried potatoes. 

#### 2.4.9. Acrylamide Content 

The acrylamide content of fried potatoes was analyzed using gas chromatography (Agilent, model GC-6890 N, Santa Clara, CA, USA) coupled with mass spectroscopy (Agilent, model MSD-5975 N, Palo Alto, CA, USA), after extraction from potatoes and derivatization with xanthydrol, following the method detailed by Molina-García et al. [18]. A DB-XLB column (Agilent, 0.25 mm I.D., 30 m length, 0.10 µm film thickness) was used for separation purposes. The carrier gas, helium, was at a constant flow of 1 mL min^−1^. The injector temperature was 250 °C, and 1 μL of sample was injected in splitless mode (pulsed pressure 32 psi, 60 s). The oven temperature was 85 °C (1 min), 18 °C min^−1^ to 280 °C, hold for 4.17 min (total of 16 min) and transfer line at 280 °C. The results were expressed in μg 100 g^−1^ of potatoes.

#### 2.4.10. Volatile Compounds

Volatile compounds of fresh and frying oils were analyzed by headspace solid-phase micro-extraction (HS-SPME) coupled with GC–MS (Agilent, Little Falls, DE, USA), according to the procedure developed by Molina-Garcia et al. [19]. An appropriate volume of internal standard (1,2,3-trichloropropane) was added to a 1.5 g mass of potato, and the mixture was heated at 50 °C for 5 min under continuous stirring. A manual SPME holder was used to expose the fiber (DVB/CAR/PDMS 50/30 μm film thickness; Supelco; Bellefonte, PA, USA) to the headspace for 30 min. The volatiles were thermally desorbed for 5 min in the injector port (270 °C, splitless) of the GC-EI-MS system. Chromatographic separation was achieved on a fused-silica SPB-5 Capillary GC column (60 m × 0.32 mm I.D. × 1 μm film thickness, Supelco) with a temperature gradient from 40 °C to 240 °C. The MS transfer line and ion source were at 250 °C, and the MS quadrupole temperature at 200 °C, with electron ionization of 70 eV, set in full scan mode (*m*/*z* 20 to 450 at 2 scan/s). Compounds were identified by comparing the respective mass spectra with a mass spectral database (WILEY7 n.L) and quantified as internal standard equivalents from their peak area ratios, expressed in μg of internal standard equivalents per 100 g^−1^ of potatoes. 

### 2.5. Statistical Analysis

All analyses were performed at least in triplicate, and results were recorded as mean ± standard deviation. Two-way analysis of variance (ANOVA) followed by multiple comparisons of averages (least significant difference (LSD) method) was chosen to show the statistical differences (*p* < 0.05) between fried potatoes in the same oil over frying time and between the fried potatoes in different oils under study using the statistical program XLSTAT (2014).

## 3. Results and Discussion

The frying assays were performed until the 15th hour of heating, when the oil level reached the minimum adapted to the fryer in use. Based on the determination of the total polar compounds in the frying oils, and as detailed in our previous study [9], all oils were below the recommended limit of 25% after 9 h of heating, as shown in Figure 1. SFO surpassed it before the 12th-hour sampling, closely followed by OO; the SFO/MOO blend broke this limit close to the 15th hour, while MOO was still below this limit even after the 15th hour of heating. Therefore, despite having analyzed potato samples up to the 15th hour of frying in some parameters, this occurred only for comparison purposes since the edibility limit based on the total polar compounds (25%) occurred earlier.

### 3.1. Color

Color perception is an important visual parameter used by consumers in the evaluation of fried products’ quality [20]. The instrumental color measurements of raw and deep-fried potatoes in different oils, both as total color changes (ΔE) and the detail of the different color coordinates (L*a*b*), are detailed in Table 1.Raw potato’s color coordinates, used as a reference, were 56.5 L* (lightness), −3.5 a* (redness) and 22.5 b* (yellowness). Although frying time and the temperature were identical for all the assays, instrumental color coordinates showed some variations, indicating that frying oils had a significant impact on the color intensities.

Globally, and in comparison with the initial color parameters of raw potatoes, deep frying potatoes promoted a decrease in coordinate L*, indicating that potatoes get darker with the frying time, and an increase in coordinates a* and b*, showing that potatoes get redder and yellower. This trend of results is in linear agreement with several authors [4,21,22].

After 15 h of frying time, potatoes fried in MOO had the higher values of L* (41.7), indicating that they were lighter, with lower values of a* (1.6) (redness) and higher values of b* (29.6) (yellowness), followed by MOO/SFO, OO and SFO. The degree of darkening was greater in the potatoes fried in SFO (L* = 35.2), slightly reduced by adding 20% of MOO to SFO (L* = 38.4), the latter equivalent to olive oil. The ΔE values are the combination of these results, with no statistical differences between fried potatoes in different oils (*p* > 0.05), except for minor but statistically significant differences after 6 h and 15 h of frying with higher ΔE values for SFO-fried potatoes and lower in MOO. The increasing ΔE trend with frying time was observed in all the oils, in agreement with Pedreschi et al. [23]. These changes in color during the frying process are the result of Maillard reactions occurring at frying temperatures, also associated with acrylamide formation [20,23,24].

### 3.2. Moisture and Fat Contents

Table 2 summarizes the moisture and fat contents of fresh and deep-fried potatoes at 3 h. The initial moisture of fresh potatoes (75.2%) showed a significant reduction, with frying to values ranging between 38.8 and 64.3%, with no specific trend over frying time or between frying oils but stabilizing around 60% after the ninth hour in all the oils. 

Raw potatoes contain small amounts of fat (0.1%) [25] (not determined in the present study), and therefore, the fat amounts of deep-fried potatoes are directly explained by fat absorption from the frying oils. The oil content of the deep-fried potatoes showed only minor differences between the studied oils (6.6–6.7%), except for SFO, which had the lowest oil absorption (5.7%). Our results for both moisture and fat contents are in line with those reported in the literature [21,22]. The water evaporation induced by the frying process creates pores in the potato structure, inevitably replaced with oil during and immediately after the frying process [4,21].

Nevertheless, this fat absorption has a nutritional impact [26], not only due to the energy balance but also due to the incorporation of nutrients and phytochemicals from frying oils into the food.

### 3.3. Fatty Acids

The amounts of saturated fatty acids (SFAs) and the main unsaturated fatty acid (UFA) classes present in fresh and deep-fried potatoes are summarized in Table 2. During deep-fat-frying operations, frying oils are absorbed by fried products and become part of the food to be consumed. Accordingly, the fatty-acid (FA) composition of fried products is not expected to differ much from the fatty-acid profile of the frying oils [21].

Oleic acid was the major FA present in potatoes fried in OO and MOO, as expected from their high-oleic composition, with no significant difference (*p* < 0.05), except at 15 h where oleic acid showed a significant decrease for potatoes fried in OO, indicative of higher fatty-acid loss, potentially by oxidation. Regarding potatoes fried in SFO, the prominent FA quantified was linoleic acid, followed by oleic acid. Blending 20% MOO with 80% SFO increased significantly (*p* < 0.05) the amounts of oleic acid in all fried potato samples. Linolenic acid was quantified in vestigial amounts for all the samples, independently of the oil. Globally, oxidation of the frying oils can be deduced in all potato samples from the decrease in UFAs through the frying time against a more stable trend for saturated ones. *Trans* fatty acids (TFAs), formed during high thermal processing and indicators of oil degradation, were present in fried potato samples in very low quantities (<0.1%), increasing slightly with frying time (Figure 2) but without statistical differences within the first 9 h of frying.

### 3.4. Vitamin E

Vitamin E constituents are among the most effective lipidic antioxidants in vegetable oils and are expected to migrate from the frying oils to the fried product because of the fat absorption during frying operations [20]. Fresh potatoes have a very limited amount of vitamin E (0.003 µg 100 g^−1^, being only represented with α-tocopherol). Therefore, the vitamin E profile of the fried food is expected to resemble the profile of the corresponding frying oils [2,21], with α-tocopherol as the most prominent vitamin E compound for all oils under study. The total vitamin E of unheated oils and deep-fried potatoes is illustrated in Table 3. At 3 h, potatoes fried in the SFO/MOO blend contained higher amounts of total vitamin E (2509 µg 100 g^−1^), followed by those fried in SFO (1098 µg 100 g^−1^), then MOO (1120 µg 100 g^−1^) and OO (401 µg 100 g^−1^) with lowest values. A significant decrease in total vitamin E was visible in all potato samples with time, as already reported by [21]. The increased content in MOO at 15 h and in OO after 9 h can potentially be explained by a slightly higher incorporation of fat [21], with 7.1% and 6.5% fat absorption for MOO and OO, respectively. 

MOO showed a richer profile in terms of vitamin E diversity, containing all tocopherols homologues together with α- and β- tocotrienols. Adding a small amount of MOO (20%) improved the vitamin E profile of SFO by more than 200%, resulting in higher total and individual vitamin E constituents. Some discussion on the degradation of these compounds with time is necessary since all the frying conditions induced severe losses in the vitamin E content. OO degradation occurred very fast, with no enrichment of the food in vitamin E already at the sixth hour of frying. SFO had a higher content of vitamin E than OO, and this was transferred proportionally to the food. The unexpectedly high content of vitamin E in the SFO/MOO blend may be due to the presence of some components that protect vitamin E constituents from degradation compared to SFO and MOO alone with frying time, as already observed in the oil [9].

### 3.5. Total Carotenoids

The carotenoid contents of fresh and deep-fried potatoes are also listed in Table 3. Fresh potatoes contained only small amounts of total carotenoids (50 µg 100 g^−1^). This value showed a significant increase (*p* < 0.05) after frying with the different oils. This phenomenon is already reported in the literature and can be justified by fat absorption during the frying process [21]. After 3 h of frying, potatoes fried in OO were richer in carotenoids, followed by those fried in MOO, the SFO/MOO blend and SFO with the lowest values. During prolonged frying, carotenoid concentrations decreased significantly (*p* < 0.05) from one sampling time to another. Here, the concentration factor imposed by water loss also contributes to an increase in potato carotenoids. Carotenoids also act as antioxidants and are probably lost during the frying process due to their action against free radicals.

The cooking process and temperature are widely reported to decrease carotenoid contents in fried potatoes during the frying process [20]. However, in some cases, and under heating conditions, carotenoids bonded to matrix proteins are dissociated, as a result of prolonged exposure to heating during cooking, allowing a detectable increase in carotenoid contents in cooked potatoes [21]. 

### 3.6. Ascorbic Acid

The high temperatures of the frying processes are regarded as an important factor in the degradation of ascorbic acid into dehydroascorbic acid, irreversibly converted into 2,3-diketogulonic acid [20,27]. Ascorbic acid amounts of fresh and deep-fried potatoes are shown in Table 4. 

Potatoes are rich sources of ascorbic acid, and the potatoes used in this study had 4.2 mg 100^−1^. Generally, ascorbic acid is significantly degraded during frying, with considerable losses (Table 4). After 3 h of frying, all fried potatoes showed an increase in vitamin C amounts, except those fried in SFO, potentially explained by the moisture decrease at this sampling. The loss in ascorbic acid concentrations is indicative of its possible contribution to the total antioxidant activity during frying, as already reported by Chu et al. [28]. Moreover, potato ascorbic acid may probably also react against free radicals formed in the frying oils during their oxidative deterioration [21], showing therefore that SFO is the most oxidized oil, whose fried potatoes contained the lowest amounts of ascorbic acid throughout the entire frying process, despite preserving vitamin E content up to the end. The effort to regenerate it through the process can potentially occur at the expense of ascorbic acid loss, which may also act as an antioxidant to scavenge free radicals present in SFO from diverse sources.

From a nutritional point of view, and taking the 9 h sampling time as a reference, the ascorbic acid amount was significantly higher in potatoes fried in high-oleic oils. A considerable improvement in the amounts of ascorbic acid in the SFO/MOO blend through all the frying time is also observed.

### 3.7. Total Phenolic Compounds and Antioxidant Activity

As detailed in Table 4, deep-fried potatoes contained higher amounts of total phenolic compounds compared to fresh ones, with only 5.9 mg GAE.100^−1^. This phenomenon can be directly explained by water loss, together with the potential enrichment from oil absorption. The phenolic compounds of deep-fried potatoes in the four oils under study showed constant to slightly increased values with frying time, which is in linear agreement with the findings of numerous studies that demonstrated the cooking process, particularly frying, to retain or even increase total phenolic contents [21,29,30], potentially released from matrix interaction due to the thermal effect.

Regarding the antioxidant activity, raw potatoes showed higher antioxidant activity (9.7 mg GAE.100 g^−1^) than fried ones in all the oils used (Table 4). However, despite the lower levels of antioxidant activity displayed by our fried potatoes, the values were consistent throughout the entire frying time. This low antioxidant activity obtained might be associated with the decrease in potato compounds with the frying time, such as ascorbic acid, carotenoids and phenolic compounds as previously discussed. Additionally, the interaction between free radicals of degraded oils present in potatoes with the potato bioactive compounds could not be excluded [21]. The slight increase in antioxidant activity despite the decrease in potato bioactive compounds with frying time agrees with the report of Kita et al. [30]. Globally, the total antioxidant activity of cooked potatoes seems to be a sum of natural antioxidants, such as carotenoids, phenolics, vitamins, etc., and the so-called new antioxidants, generated from caramelization, Maillard reaction, Strecker degradation, hydrolysis of esters and glycosides of antioxidants and oxidation of phenolic antioxidants to quinones and their polymers [31]. 

### 3.8. Acrylamide

Acrylamide in cooked foodstuffs has caused a worldwide burden, since it is classified as a carcinogenic molecule for humans, with potato chips and French fries being among the cooked foods containing high concentrations [32]. In this study, acrylamide contents were determined only on some selected samples (after 3 h and 9 h heating time), taking the 25% of total polar compounds for all oils as guidance.

Acrylamide contents of deep-fried potatoes in the studied oils are shown in Table 4 and are within the indicated benchmark levels for the main foodstuff contributors to acrylamide exposure by the European Commission in 2017 [33], with 50 µg 100 g^−1^ being the reference settled for French fries, except for deep-fried potatoes in SFO (103µg 100 g^−1^) and OO (60 µg 100 g^−1^) after 3 h and 6 h of frying, respectively. Potatoes fried in MOO showed the lowest concentrations, and those fried in SFO/MOO blend showed lower acrylamide values compared to those fried in 100% SFO. Acrylamide formation is linked to the initial precursors available in raw potatoes such as reducing sugars, which should be very low in fresh potatoes intended for French fries’ preparation [22]. Additionally, the temperature reached in the frying processes, required to attain the desirable color, flavor and aroma production, as well as the processing time, among other factors, is associated with acrylamide formation [24,34]. Knowing that the frying temperature and potato variety were the same among all the frying experiments, the observed differences can be attributed mostly to the type of frying oil used, involving fatty acids and the antioxidant pool present in the oil, without excluding variation due to the doneness degree [32].

### 3.9. Volatile Compounds

The volatile compounds of the deep-fried potatoes in different oils under study were compared only after 9 h of heating time, being the limit point linked to human health safety (TPC < 25% for the oils studied), and were only used to tentatively perceive the chemical formation of compounds resulting from fatty-acid oxidation. The amounts of the different volatile compounds, grouped by chemical families, are detailed in Table 5, grouped as non-aldehydes and aldehydes volatiles. Deep-fried potatoes in OO and SFO showed similar amounts of total volatiles, both significantly higher than those fried in the SFO/MOO blend and MOO (*p* < 0.05).

This indicates that the type of frying oil had an impact on the volatile compound formation, and also that blending SFO with 20% MOO reduced the total amounts of volatile compounds by about 53%. Moreover, the contribution of the aldehydes to the total volatiles was higher in the polyunsaturated oils (SFO > SFO/MOO > OO > MOO). Alkanals and alkenals were the most prominent aldehyde constituents in OO (2-undecenal > nonanal) and MOO (nonanal > 2-undecenal), resulting both from oleic-acid oxidation. Alkadienals, mainly represented by EE-2,4-decadienal (Figure 3), were higher in SFO and the SFO/MOO blend, explained by their higher amounts of linoleic acid [32]. Potatoes fried in MOO were exempt from alkadienals and contained very low levels of alkenals, both recognized for their increased toxicity for humans [34]. As previously demonstrated for the vitamin E content, the amount of aldehydes in the potatoes fried in the SFO/MOO blend is also lower than could be expected from their mass balances. Using only 20% of MOO decreased linoleic acid proportionally but resulted in a reduction of 40% in the aldehyde contents, a probable consequence of a lower oxidation state of the blend.

As to the non-aldehyde compounds, pyrazines class was the main class present in fried potatoes in OO and SFO, while present in low amounts in the other oils. These compounds are known to result mostly from sugar/Maillard reactions degradation from the potato constituents [21]. 

## 4. Conclusions

Raw potatoes are rich in bioactive compounds such as vitamin C, carotenoids and phenolic compounds known to act as radical scavengers against lipid oxidation. During frying, fried potatoes are enriched in oil compounds, while the potatoes’ own and acquired bioactive compounds are degraded and lost during frying time, reducing therefore the nutritive value of fried potatoes. Simultaneously, fried potatoes are enriched in newly formed compounds (TFAs, acrylamides and volatile aldehydes resulting from fatty-acid oxidation) whose intake should be limited.

This study showed a clear influence of the vegetable oil composition features on the potato’s composition. Noticeable losses were observed in the fried potato constituents and the antioxidant activity as a result of both potato and oil antioxidant degradation. TFAs, acrylamide and volatile compounds are formed and incorporated in very low concentrations. 

As to the relevance of using MOO, potatoes fried in MOO had lower concentrations of newly formed compounds and a more interesting nutritional profile than all the remaining ones. Moreover, blending 20% MOO with SFO significantly improved the quality of fried potatoes compared to those fried in SFO alone.

## Figures and Tables

**Figure 1 foods-12-04416-f001:**
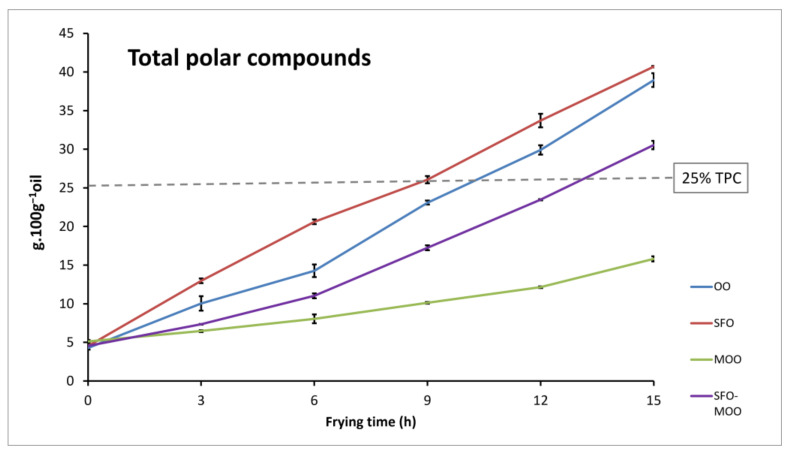
Total polar compounds in the oil (g 100 g^−1^) during the frying time.

**Figure 2 foods-12-04416-f002:**
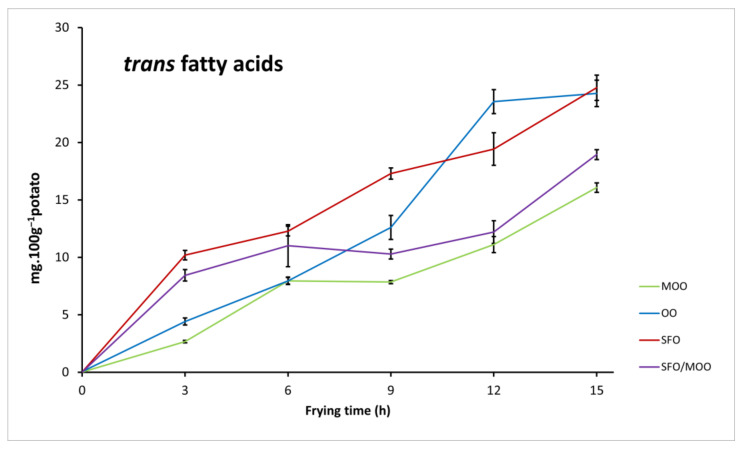
*Trans* fatty acids in deep-fried potatoes in MOO, OO, SFO and MOO/SFO blend overtime.

**Figure 3 foods-12-04416-f003:**
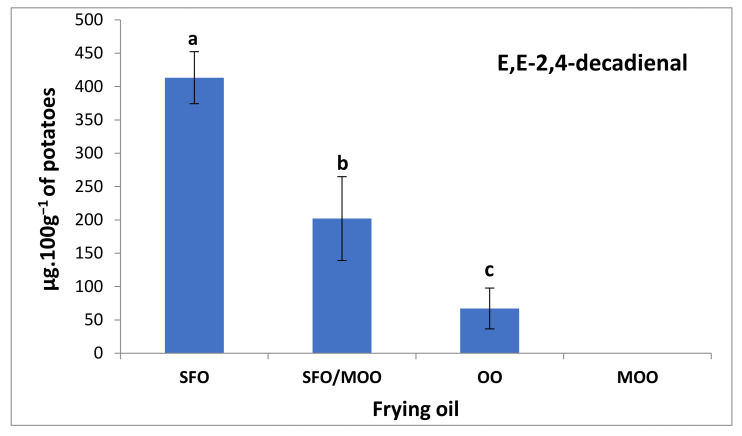
Differences in E,E-2,4-decadienal amounts in deep-fried potatoes after 9 h of heating. Letters a–c indicate a significant difference between potatoes fried on different vegetable oils.

**Table 1 foods-12-04416-t001:** Color variation of intermittent deep-fried potatoes in MOO, OO, SFO and MOO/SFO blend.

			Potatoes Fried At				
		Raw Potato	3 h	6 h	9 h	12 h	15 h
L*	MOO	56.5 ± 1.0	46.0 ± 1.0 ^a,A^	44.7 ± 0.6 ^b,A^	42.3 ± 18.9 ^c,A^	41.7 ± 2.0 ^d,A^	41.7± 0.4 ^d,A^
	OO		44.2 ± 0.3 ^a,B^	41. 0± 1.2 ^b,C^	41.1 ± 17.3 ^b,B^	38.7 ± 1.1 ^c,C^	38.0± 0.4 ^d,B^
	SFO		42.8 ± 0.7 ^a,C^	40.9 ± 0.2 ^b,D^	40.6 ± 1.7 ^b,C^	38.3 ± 0.8 ^c,C^	35.2 ± 1.6 ^d,C^
	MOO/SFO		44.6 ± 3.7 ^a,B^	43.8 ± 0.3 ^b,B^	41.5 ± 1.7 ^c,B^	39.6 ± 1.1 ^d,B^	38.4 ± 0.7 ^e,B^
a*	MOO	−3.5 ± 0.1	0.5 ± 0.3 ^c,A^	1.2 ± 0.4 ^b,A^	1.2 ± 1.1 ^b,A^	1.5 ± 0.4 ^a,B^	1.6± 0.5 ^a,C^
	OO		−0.8 ± 1.2 ^e,B^	0.6 ± 0.4 ^d,B^	1.2 ± 1.7 ^c,A^	1.7 ± 0.3 ^b,A^	2.1 ± 0.3 ^a,A^
	SFO		−1.6 ± 1.4 ^e,C^	0.2 ± 1.3 ^d,BC^	1.0 ± 1.1 ^c,B^	1.6 ± 0.6 ^b,A^	2.0 ± 1.0 ^a,B^
	MOO/SFO		−1.0 ± 0.4 ^d,B^	-0.3 ± 1.0 ^c,C^	1.2 ± 1.2 ^b,A^	1.3 ± 0.1 ^b,C^	1.6 ± 0.4 ^a,C^
b*	MOO	22.5 ± 0.1	23.5 ± 0.2 ^e,B^	25.0 ± 0.8 ^d,A^	26.1 ± 1.4 ^bc,A^	27.1 ± 3.1 ^a,A^	29.6 ± 1.4 ^a,A^
	OO		21.5 ± 0.8 ^d,D^	23.0 ± 1.2 ^c,C^	23.4 ± 0.9 ^c,C^	24.9 ± 0.5 ^b,C^	25.5 ± 0.5 ^a,C^
	SFO		22.1 ± 0.4 ^e,C^	22.7 ± 0.4 ^d,D^	23.8 ± 0.8 ^c,C^	24.1 ± 0.1 ^b,D^	24.9 ± 0.6 ^a,D^
	MOO/SFO		24.2 ± 0.6 ^c,A^	24.4 ± 0.9 ^c,B^	25.0 ± 0.9 ^b,B^	25.7 ± 1.1 ^b,B^	27.2 ± 0.9 ^a,B^
ΔE	MOO	-	9.8 ^c,A^	11.1 ^bc,C^	13.8 ^bc,A^	14.7 ^ab,A^	15.6 ^a,B^
	OO	-	12.1 ^c,A^	14.9 ^bc,AB^	14.7 ^bc,A^	17.2 ^ab,A^	17.9 ^a,AB^
	SFO	-	13.6 ^c,A^	15.2 ^bc,A^	15.3 ^bc,A^	17.6 ^b,A^	20.7 ^a,A^
	MOO/SFO	-	11.7 ^b,A^	12.4 ^b,BC^	14.5 ^ab,A^	16.5 ^ab,A^	18.0 ^a,B^

Small letters indicate a significant difference between deep-fried potatoes over frying time, and capital letters between oils.

**Table 2 foods-12-04416-t002:** Moisture content, fat content * and fatty-acid composition of intermittent deep-fried potatoes in MOO, OO, SFO and MOO/SFO blend.

		Raw Potato Unheated	Potatoes Fried At				
g 100 g^−1^	Oils	0 h	3 h	6 h	9 h	12 h	15 h
Moisture	MOO	75.2 ± 0.1	50.6 ± 1.1 ^a,B^	59.2 ± 8.3 ^a,A^	61.8 ± 2.5 ^a,A^	61.8 ± 1.2 ^a,A^	51.6 ± 2.7 ^c,B^
	OO		38.8 ± 1.3 ^b,C^	51.1 ± 2.5 ^a,B^	59.9 ± 5.9 ^a,A^	58.3 ± 1.6 ^b,A^	59.6 ± 0.9 ^b,A^
	SFO		51.0 ± 1.01 ^a,C^	60.6 ± 0.8 ^a,B^	60.6 ± 0.7 ^a,B^	61.5 ± 1.1 ^a,B^	64.3 ± 0.2 ^a,A^
	MOO/SFO		45.6 ± 6.4 ^a,B^	55.3 ± 7.7 ^a,A^	59.3 ± 1.4 ^a,A^	60.2 ± 2.0 ^a,A^	61.0 ± 1.7 ^b,A^
Fat Content *	MOO	ND	6.6 ± 0.6 ^a^				
	OO		6.7 ± 0.5 ^a^				
	SFO		5.7 ± 0.5 ^b^				
	MOO/SFO		6.6 ± 0.3 ^a^				
SFA	MOO	0.23 ± 0.01 ^a,B^	1.2 ± 0.0 ^c,A^	1.6 ± 0.1 ^a,A^	1.4 ± 0.0 ^b,A^	1.4 ± 0.0 ^b,A^	1.4 ± 0.0 ^b,A^
	OO	0.16 ± 0.01 ^b,A^	1.1 ± 0.0 ^b,B^	1.1 ± 0.0 ^ab,B^	0.9 ± 0.1 ^c,B^	1.2 ± 0.0 ^a,B^	0.9 ± 0.1 ^c,B^
	SFO	0.10 ± 0.00 ^d,B^	0.6 ± 0.0 ^b,D^	0.6 ± 0.0 ^b,D^	0.7 ± 0.0 ^a,C^	0.6 ± 0.0 ^a,D^	0.7 ± 0.0 ^a,C^
	MOO/SFO	0.14 ± 0.01 ^c,A^	0.8 ± 0.1 ^b,C^	0.8 ± 0.0 ^bc,C^	0.7 ± 0.0 ^d,C^	0.7 ± 0.0 ^cd,C^	0.9 ± 0.0 ^a,B^
C18:1	MOO	0.70 ± 0.01 ^b,A^	4.4 ± 0.0 ^ab,A^	4.6 ± 0.3 ^a,A^	3.9 ± 0.1 ^c,A^	3.7 ± 0.2 ^c,B^	3.9 ± 0.1 ^bc,A^
	OO	0.72 ± 0.00 ^a,A^	4.5 ± 0.2 ^a,A^	4.7 ± 0.1 ^a,A^	3.8 ± 0.3 ^b,A^	4.5 ± 0.1 ^a,A^	3.4 ± 0.2 ^b,B^
	SFO	0.32 ± 0.00 ^d,B^	1.7 ± 0.0 ^a,C^	1.7 ± 0.0 ^b,C^	1.9 ± 0.1 ^b,B^	1.8 ± 0.0 ^a,C^	1.9 ± 0.0 ^a,D^
	MOO/SFO	0.38 ± 0.01 ^c,B^	2.3 ± 0.2 ^b,B^	2.2 ± 0.0 ^b,B^	1.9 ± 0.0 ^c,B^	2.0 ± 0.0 ^c,C^	2.5 ± 0.1 ^a,C^
C18:2	MOO	0.01 ± 0.00 ^d,A^	0.1 ± 0.0 ^a,D^	0.1 ± 0.0 ^a,D^	0.1 ± 0.0 ^a,D^	0.1 ± 0.0 ^a,D^	0.1 ± 0.0 ^a,D^
	OO	0.76 ± 0.01 ^c,A^	0.5 ± 0.0 ^a,C^	0.5 ± 0.0 ^a,C^	0.4 ± 0.0 ^b,C^	0.4 ± 0.0 ^b,C^	0.3 ± 0.0 ^c,C^
	SFO	0.54 ± 0.01 ^b,A^	2.8 ± 0.1 ^a,A^	2.6 ± 0.1 ^b,A^	2.8 ± 0.1 ^a,A^	2.6 ± 0.0 ^b,A^	2.5 ± 0.1 ^b,A^
	MOO/SFO	0.43 ± 0.00 ^a,A^	2.6 ± 0.2 ^a,B^	2.5 ± 0.0 ^a,B^	2.1 ± 0.0 ^b,B^	2.0 ± 0.0 ^b,B^	2.4 ± 0.1 ^a,B^
C18:3	MOO	0.002 ± 0.001 ^b,A^	0.02 ± 0.00 ^a,A^	0.02 ± 0.00 ^a,B^	0.02 ± 0.00 ^a,B^	0.02 ± 0.0 ^a,AB^	0.02 ± 0.00 ^a,A^
	OO	0.006 ± 0.001 ^a,A^	0.05 ± 0.00 ^a,A^	0.04 ± 0.00 ^b,A^	0.03 ± 0.00 ^c,A^	0.03 ± 0.00 ^c,A^	0.02 ± 0.00 ^a,A^
	SFO	0.001 ± 0.001 ^d,A^	0.02 ± 0.00 ^a,A^	0.01 ± 0.00 ^a,C^	0.02 ± 0.00 ^a,B^	0.02 ± 0.0 ^a,BC^	0.02 ± 0.00 ^a,A^
	MOO/SFO	0.001 ± 0.001 ^c,A^	0.02 ± 0.00 ^a,A^	0.02 ± 0.0 ^ab,B^	0.01 ± 0.00 ^b,C^	0.01 ± 0.0 ^ab,C^	0.02 ± 0.00 ^a,A^

* Fat content is given as the mean of all fat content determinations of 15 h frying. Small letters indicate a significant difference between deep-fried potatoes over frying time, and capital letters between oils. ND: not determined.

**Table 3 foods-12-04416-t003:** Vitamin E constituents and total carotenoids changes in unheated oils (mg 100 g^−1^) and in intermittent deep-fried potatoes (µg100 g^−1^) in MOO, OO, SFO and MOO/SFO blend.

		Unheated Oils	Potatoes Fried At				
µg 100 g^−1^		(mg 100 g^−1^)	3 h	6 h	9 h	12 h	15 h
Total Vitamin E	MOO	29.0 ± 1.6 ^b,A^	1120 ± 2 ^a,C^	450 ± 0 ^b,C^	144 ± 0 ^c,C^	57 ± 0 ^d,D^	86 ± 0 ^e,C^
	OO	15.4 ± 0.0 ^c^	401 ± 1 ^a,D^	38 ± 0 ^d,D^	57 ± 0 ^c,D^	102 ± 0 ^b,C^	58 ± 0 ^c,D^
	SFO	49.2 ± 1.2 ^a,A^	1098 ± 5 ^a,B^	1191 ± 1 ^a,B^	497 ± 0 ^b,B^	149 ± 1 ^c,B^	116 ± 1 ^d,B^
	MOO/SFO	48.1 ± 0.5 ^a,A^	2509 ± 2 ^a,A^	2241 ± 2 ^b,A^	1162 ± 1 ^c,A^	510 ± 0 ^d,A^	147 ± 0 ^e,A^
α-Tocopherol	MOO	19.0 ± 1.0 ^c,A^	782 ± 1 ^a,AB^	272 ± 0 ^b,C^	105 ± 0 ^c,C^	42 ± 0 ^e,D^	86 ± 0 ^d,B^
	OO	14.2 ± 0.4 ^d^	355 ± 1 ^a,B^	31 ± 0 ^d,D^	50 ± 0 ^c,D^	91 ± 0 ^b,C^	50 ± 0 ^c,C^
	SFO	46.9 ± 1.1 ^a,A^	1038 ± 5 ^a,AB^	1130 ± 1 ^a,B^	436 ± 0 ^b,B^	117 ± 0 ^c,B^	92 ± 1 ^c,A^
	MOO/SFO	43.6 ± 0.4 ^b,A^	2315 ± 1 ^a,A^	2076 ± 2 ^a,A^	1084 ± 1 ^ab,A^	470 ± 0 ^b,A^	125 ± 0 ^b,A^
β-Tocopherol	MOO	1.08 ± 0.6 ^b,A^	45.0 ± 0.1 ^a,C^	30.0 ± 0.0 ^b,C^	10.0 ± 0.0 ^c,C^	5.2 ± 0.0 ^d,B^	ND
	OO	0.18 ± 0.0 ^c^	19.0 ± 0.1 ^D^	ND	ND	ND	ND
	SFO	1.85 ± 0.07 ^a,A^	50.2 ± 0.7 ^a,B^	51.4 ± 0.5 ^b,B^	57.3 ± 0.1 ^ab,A^	32.2 ± 0.1 ^c,A^	23.6 ± 0.1 ^d,A^
	MOO/SFO	1.76 ± 0.04 ^a,A^	95.1 ± 0.5 ^a,A^	90.4 ± 0.1 ^a,A^	57.1 ± 0.0 ^b,B^	35.3 ± 0.1 ^c,A^	21.8 ± 0.0 ^d,B^
δ-Tocopherol	MOO	0.73 ± 0.05 ^a,A^	26.6 ± 0.1 ^a,A^	29.6 ± 0.0 ^b^	16.9 ± 0.0 ^c^	9.9 ± 0.0 ^d^	ND
	OO	0.61 ± 0.02 ^b^	16.8 ± 0.1 ^B^	ND	ND	ND	ND
	SFO	ND	ND	ND	ND	ND	ND
	MOO/SFO	0.14 ± 0.01 ^c,A^	ND	ND	ND	ND	ND
γ-Tocopherol	MOO	6.56 ± 0.36 ^a,A^	242.1 ± 0.2 ^a,A^	104.2 ± 0.2 ^b,A^	11.2 ± 0.0 ^c,B^	ND	ND
	OO	0.69 ± 0.02 ^c^	10.7 ± 0.1 ^a,C^	7.5 ± 0.0 ^bc,D^	7.6 ± 0.0 ^c,C^	10.6 ± 0.0 ^ab,A^	7.8 ± 0.1 ^c^
	SFO	0.45 ± 0.04 ^c,B^	10.1 ± 0.2 ^a,C^	9.5 ± 0.1 ^a,C^	3.4 ± 0.0 ^b,C^	ND	ND
	MOO/SFO	1.66 ± 0.04 ^b,A^	74.1 ± 0.3 ^a,B^	56.9 ± 0.1 ^b,B^	20.6 ± 0.0 ^c,A^	4.3 ± 0.0 ^d,B^	ND
α-Tocotrienol	MOO	0.56 ± 0.03 ^c^	18.2 ± 0.2 ^a^	3.5 ± 0.0 ^b^	ND	ND	ND
	OO	ND	ND	ND	ND	ND	ND
	SFO	ND	ND	ND	ND	ND	ND
	MOO/SFO	0.28 ± 0.01	ND	ND	ND	ND	ND
β- Tocotrienol	MOO	1.07 ± 0.05 ^a,A^	35.3 ± 0.1 ^a,A^	10.5 ± 0.0 ^b,B^	ND	ND	ND
	OO	ND	ND	ND	ND	ND	ND
	SFO	ND	ND	ND	ND	ND	ND
	MOO/SFO	0.58 ± 0.01 ^b,A^	24.8 ± 0.2 ^a,B^	17.8 ± 0.1 ^b,A^	ND	ND	ND
Total carotenoids	MOO	0.50 ± 0.00 ^b,A^	211.9 ± 0.8 ^a,B^	203.4 ± 1.6 ^b,A^	150.0 ± 1.6 ^c,B^	133.9 ± 2.1 ^d,B^	132.0 ± 2.9 ^d,A^
	OO	0.77 ± 0.04 ^a,A^	233.5 ± 0.7 ^a,A^	201.3 ± 1.4 ^b,A^	162.9 ± 1.2 ^c,A^	152.2 ± 2.2 ^d,A^	134.4 ± 1.6 ^e,A^
	SFO	0.17 ± 0.00 ^c,A^	164.9 ± 5.0 ^a,D^	145.4 ± 2.9 ^b,C^	130.6 ± 1.1 ^c,C^	128.4 ± 6.6 ^c,C^	122.4 ± 0.6 ^c,B^
	MOO/SFO	0.19 ± 0.01 ^c,AB^	172.6 ± 13.5 ^a,C^	158.8 ± 2.1 ^ab,B^	147.8 ± 4.2 ^bc,B^	141.1 ± 1.4 ^cd,BC^	130.8 ± 2.0 ^d,A^

Small letters indicate a significant difference between deep-fried potatoes over frying time, and capital letters between oils. ND: not determined.

**Table 4 foods-12-04416-t004:** Changes in composition and antioxidant capacity of potatoes during intermittent deepfrying in MOO, OO, SFO and MOO/SFO blend.

	Oils Name	Potatoes	Fried At				
		Raw	3 h	6 h	9 h	12 h	15 h
Ascorbic acid	MOO	4.2 ± 0.1	6.4 ± 0.0 ^b,A^	6.8 ± 0.1 ^a,A^	5.6 ± 0.1 ^c,A^	1.9 ± 0.1 ^d,A^	4.0 ± 0.1 ^e,A^
(mg 100 g^−1^)	OO		4.6 ± 0.5 ^b,B^	4.1 ± 0.1 ^bc,C^	5.4 ± 0.1 ^a,A^	4.3 ± 0.2 ^b,B^	3.7 ± 0.2 ^c,A^
	SFO		2.0 ± 0.3 ^b,C^	3.7 ± 0.4 ^a,C^	3.6 ± 0.4 ^a,C^	3.7± 0.2 ^a,C^	2.7 ± 0.1 ^b,B^
	MOO/SFO		6.2 ± 0.2 ^a,D^	5.8 ± 0.1 ^a,B^	4.6 ± 0.3 ^c,B^	5.1 ± 0.1 ^b,A^	4.2 ± 0.3 ^c,B^
Total phenolics	MOO	5.9 ± 0.9	8.1 ± 18.2 ^a,A^	8.0 ± 5.8 ^a,AB^	7.8 ± 1.5 ^a,B^	8.2 ± 5.6 ^a,AB^	8.9 ± 6.5 ^a,AB^
(mg GAE.100 g^−1^)	OO		8.3 ± 4.3 ^b,A^	8.8 ± 5.1 ^ab,A^	9.2 ± 3.4 ^ab,A^	9.6 ± 4.2 ^a,A^	9.8 ± 8.7 ^a,A^
	SFO		8.7 ± 1.3 ^a,A^	8.1 ± 6.4 ^ab,AB^	6.8 ± 3.7 ^c,C^	8.1 ± 6.9 ^ab,BC^	7.5 ± 6.0 ^bc,B^
	MOO/SFO		7.8 ± 14.9 ^ab,A^	7.8 ± 2.4 ^ab,B^	9.2 ± 1.4 ^a,A^	7.2 ± 4.4 ^b,C^	9.0 ± 3.4 ^a,A^
DPPH assay	MOO	9.7 ± 0.7	2.2 ± 7.8 ^a,A^	2.3 ± 8.6 ^a,A^	1.9 ± 8.8 ^a,A^	3.0 ± 4.4 ^a,A^	3.0 ± 25.3 ^a,A^
(mg GAE.100 g^−1^)	OO		2.0 ± 8.7 ^a,A^	1.7 ± 12.1 ^a,A^	2.5 ± 0.0 ^a,A^	2.5± 8.3 ^a,A^	2.1 ± 11.3 ^a,A^
	SFO		2.7 ± 10.5 ^a,A^	1.3 ± 11.7 ^c,A^	1.6 ± 7.9 ^c,A^	1.8 ± 7.7 ^bc,A^	2.5 ± 5.4 ^ab,A^
	MOO/SFO		1.8 ± 2.6 ^a,A^	1.8 ± 4.1 ^a,A^	1.6 ± 3.3 ^a,A^	2.0 ± 3.2 ^a,A^	1.6 ± 1.3 ^a,A^
Acrylamide	MOO	-	15 ± 28 ^c,C^	-	28 ± 68 ^b,C^		
(µg 100 g^−1^)	OO		41 ± 68 ^b,B^	-	60 ± 31 ^a,A^		
	SFO		103 ± 87 ^a,A^	-	50 ± 33 ^b,AB^		
	MOO/SFO		41 ± 17 ^a,B^	-	40 ± 27 ^a,BC^		

Small letters indicate a significant difference between deep-fried potatoes over frying time, and capital letters between oils.

**Table 5 foods-12-04416-t005:** Changes in volatile compounds in deep-fried potatoes after 9 h heating time (in µg internal standard equivalents 100 g^−1^).

Volatiles Families	MOO	OO	SFO	SFO/MOO
Alcohols	ND	19 ± 2 ^A^	21 ± 2 ^A^	7 ± 4 ^B^
Alkylbenzenes	49 ± 2 ^A^	51 ± 13 ^A^	19 ± 5 ^B^	48 ± 15 ^A^
Hydrocarbons	6 ± 1 ^B^	25 ± 5 ^A^	6 ± 0 ^B^	ND
Carboxylic acids	20 ± 23 ^B^	ND	43 ± 10 ^A^	ND
Ketones	ND	6 ± 0 ^B^	48 ± 3 ^A^	ND
Furan derivates	16 ± 2 ^B^	19 ± 2 ^B^	40 ± 3 ^A^	24 ± 7 ^B^
Pyrazines	25 ± 5 ^C^	252 ± 93 ^A^	115 ± 6 ^B^	40 ± 10 ^C^
Total non-aldehydes	117 ± 16 ^B^	372 ± 104 ^A^	289 ± 21 ^A^	119 ± 36 ^B^
Alkanals	202 ± 41 ^B^	287 ± 159 ^A^	82 ± 14 ^D^	137 ± 24 ^C^
Alkenals	63 ± 17 ^B^	199 ± 68 ^A^	119 ± 12 ^AB^	59 ± 23 ^B^
Alkadienals	ND	77 ± 29 ^C^	436 ± 38 ^A^	182 ± 59 ^B^
Total aldehydes	265 ± 98 ^B^	363 ± 214 ^AB^	637 ± 64 ^A^	377 ± 80 ^AB^
Total volatiles	383 ± 68 ^B^	935 ± 216 ^A^	926 ± 84 ^A^	497 ± 115 ^B^

Letters A–D indicate a significant difference between vegetable oils. ND: not detected.

## Data Availability

Data are contained within the article.

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
