# Peer review of "Moringa oleifera Oil Nutritional and Safety Impact on Deep-Fried Potatoes"

_foods, 2023, doi:10.3390/foods12244416_

Round 1
Reviewer 1 Report
Comments and Suggestions for Authors
- Clarity and Structure:
- The manuscript lacks an introduction or abstract to provide context for the reader. Consider starting with a brief introduction to the topic and the objectives of your study.
- It's important to clearly state the research question or hypothesis that your study aims to address. This will help the reader understand the purpose of your work.
- Methodology:
- Provide more details about the sample preparation and analysis methods. Explain how the potato samples were prepared for analysis and the specific techniques used to measure compounds (e.g., mass spectrometry, color measurements).
- Specify the type of potatoes used in the study, as different potato varieties can have varying compositions.
- Describe the instruments and software used for analysis in more detail. Include relevant parameters or settings if applicable.
- When referring to figures, it's common to mention the figure number (e.g., "as shown in Figure 1") to help readers locate the visual data.
- Results and Discussion:
- In the results section, present the data in a more organized and reader-friendly manner. Tables and figures can be used to convey data more efficiently.
- Discuss the findings in more detail. Explain the implications of the results and how they relate to the research question.
- Consider providing a summary of the key findings at the end of this section to highlight the most important results.
- Statistical Analysis:
- Provide more information about the statistical methods used. What type of statistical tests were employed, and why were they chosen?
- References:
- Make sure to include proper citations and references. For example, reference [12] is mentioned, but the actual citation is missing. Ensure that all sources are cited appropriately.
- Nomenclature and Abbreviations:
- Define any acronyms or abbreviations used in the manuscript, especially when they are first introduced.
- Consider creating a nomenclature section to explain the meaning of terms specific to your study.
- Visuals:
- If the manuscript includes figures, provide captions that explain what each figure represents. Make sure that figure legends are placed beneath the figures for clarity.
- Language and Style:
- Ensure consistency in formatting, especially with respect to units (e.g., μg, g) and references (e.g., [12]).
- Proofread the manuscript to correct any grammatical or typographical errors.
- Conclusion:
- The manuscript excerpt provided does not include a conclusion section. Be sure to include a conclusion that summarizes the key findings and their significance.
no issues
Author Response
Dear Reviewer,
Hope this message finds you well!
Please see the attachment!
Best regards
Silia Boukandoul

Reviewer 2 Report
Comments and Suggestions for Authors
The research manuscript is of interest to researchers and also nutritionists. However the English language needs to be revised so that the readability of the manuscript is improved. For instance the abstract needs to be revised so that it very clearly shows the significance of the research in the results obtained. The introduction needs reformatting to improve the flow of sentences and also to add deeper critical analysis in terms of the use of Moringa oleifera oil and other oils in food systems and their biological activities, comparing and contrasting the current research with that of previous researchers, for instance:
Karami, Z., Akbari-adergani, B. and Duangmal, K. (2022), Recent development on recovering bioactive peptides and phenolic compounds from under-utilised by-products during production of certain edible oil plants: current situation and future perspectives. Int J Food Sci Technol, 57: 4894-4905. https://doi.org/10.1111/ijfs.15838
It would also be of interest to have a discussion about the frying process in general so as to understand the nutritional and functional aspects relevant to frying and processing conditions due to heat and other aspects. Many of the references are old and do not cover research which has occurred in the last 3-4 years. It would be useful if the authors could include some new research such as:
Sosa-Morales, M.E., Solares-Alvarado, A.P., Aguilera-Bocanegra, S.P., Muñoz-Roa, J.F. and Cardoso-Ugarte, G.A. (2022), Reviewing the effects of vacuum frying on frying medium and fried foods properties. Int J Food Sci Technol, 57: 3278-3291. https://doi.org/10.1111/ijfs.15572
Figure 1 and 2 need to be a little clearer, and is it possible for the authors to add letters to the specific points indicating statistical significance over time of processing ?
Table 2 needs also to be reedited so that the errors and data points are easy to follow within the table. This will help with the understanding of the significant variations and where they occur. Could the authors expand on the mechanisms of the differences in colour and how processing and oil components have altered colour in the products?
Why was acrylamide only tested in some of the products and not others ? How do your results compare with acrylamide formation in other processing conditions and different oils used by other researchers ? Please expand on why this is important and why this plays a role in human nutrition and gut functionality .
Overall the manuscript is useful and provides some relevant information. However the literature cited is old (the most recent publication is 4-5 years old) and also the English needs to be improved in the Abstract and Introduction sections so that there is a clarity in the presentation.
Author Response

(The authors gave the same response as above.)
